# “Orphaned” Stomach—An Infrequent Complication of Gastric Bypass Revision

**DOI:** 10.3390/jcm11247487

**Published:** 2022-12-17

**Authors:** Dimitrios N. Varvoglis, Manuel Sanchez-Casalongue, Todd H. Baron, Timothy M. Farrell

**Affiliations:** 1Department of Surgery, University of North Carolina at Chapel Hill, Chapel Hill, NC 27514, USA; 2Department of Medicine, University of North Carolina at Chapel Hill, Chapel Hill, NC 27514, USA

**Keywords:** excluded stomach, orphaned stomach, gastric bypass revision, trans-gastric stenting

## Abstract

While generally safe, bariatric operations have a variety of possible complications. We present an uncommon complication after gastric bypass revision, namely the creation of an “orphaned” segment of remnant stomach that was left inadvertently in discontinuity, leading to recurrent intra-abdominal abscesses. Sinogram ultimately proved the diagnosis, and the issue was successfully treated using a combination of surgical and endoscopic methods to control the abscess and to allow internal drainage.

## 1. Introduction

Obesity is a major public health problem worldwide. It affects quality of life and contributes to many comorbid conditions, including type-II diabetes, hypertension, hyperlipidemia, coronary artery disease, and several cancers [1]. Non-surgical treatments include behavior modification through diet, exercise, and psychological support, as well as pharmacotherapy. Unfortunately, many patients fail to respond adequately.

Bariatric surgery offers the best contemporary option for effective and durable weight loss. Different bariatric procedures exist, including restrictive operations such as sleeve gastrectomy and adjustable gastric banding, and combined restrictive and malabsorptive operations such as Roux-en-Y gastric bypass (RGB) and duodenal switch. According to recent data, RGB is the second most commonly performed bariatric operation [2]. RGB has long been considered the “gold standard” against which newer operations are compared. Even though sleeve gastrectomy is presently more popular worldwide, RGB is still commonly utilized, especially for patients with gastroesophageal reflux and sweet-eating tendencies. In fact, as recently as 2014, RGB was the principle bariatric operation performed in the United States [3].

While the expected excess weight loss after RGB ranges from 60–75% at 5 years [4,5,6,7], more than half of patients may have inadequate weight loss or substantial regain of weight [8,9]. Likewise, some RGB patients may present with late anatomic complications such as recurrent marginal ulcer, gastrojejunal stricture, or gastro-gastric fistula [10,11,12,13,14,15,16,17,18]. Therefore, surgeons are likely to encounter patients with RGB anatomy who request revisional procedures for these reasons.

## 2. Detailed Case Description

A 43-year-old woman presented in mid-2020 after a complex past surgical history. She initially had a laparoscopic Roux-en-Y gastric bypass in 2014 for severe obesity, but she had a regain of weight. In late 2019, she had laparoscopic revision, which included reduction of her gastric pouch and redo of her gastrojejunostomy. In doing this, the surgeon resected a portion of her remnant stomach that was densely adherent to the posterior gastrojejunostomy.

The patient had a complicated postoperative course and developed intra-abdominal abscesses which were not readily amenable to percutaneous drainage. Therefore, in early 2020, she underwent laparoscopy, which required conversion to open laparotomy for an abdominal washout. At operation, the presumed left upper quadrant fluid collection was judged to be a segment of the remnant stomach.

Shortly thereafter, the patient moved to North Carolina and was admitted at a small hospital complaining of abdominal and left shoulder pain. On computerized tomography (CT) scan, a perisplenic fluid collection was noted (Figure 1). She was admitted and underwent percutaneous drain placement by interventional radiology. Despite intravenous antibiotics, the patient developed a significant leukocytosis with increasing drain output and was transferred to our academic medical center for further management.

The patient underwent a sinogram that showed the drain was in an apparent abscess cavity. An upper gastrointestinal contrast study (Figure 2) and CT sinogram both failed to show evidence of a fistula feeding the cavity. The patient had resolution of her leukocytosis and was discharged to home with the drain in place.

After three weeks, a follow-up sinogram showed no residual cavity, and the drain was removed. However, the patient was admitted shortly thereafter due to severe pain. A repeat CT scan (Figure 3) showed a multiloculated fluid collection above the remnant stomach that had been separated from the previous drain.

The patient was taken to operation and had laparoscopic lysis of adhesions and washout of a purulent cavity at the above-noted site. A surgical drain was left in place (Figure 4). A remnant gastrostomy was added for decompression of the distal stomach. Her symptoms improved dramatically. She was sent home on broad-spectrum IV antibiotics with closed suction drainage of her abscess cavity and gravity drainage of her remnant stomach. The antibiotic regimen was later refined to Augmentin and Fluconazole based on cultures.

The patient was evaluated in clinic 10 days later, at which time her remnant gastrostomy was clamped due to low output. She returned two weeks after that, and a repeat sinogram showed a small fistula between a portion of the remnant stomach and the surgical drain (Figure 5). This area did not connect with the proximal gastric pouch or the alimentary limb.

About six weeks after her most recent operation, she had retrograde endoscopy through the remnant gastrostomy site, which demonstrated no connection into the “orphaned” portion of stomach that was feeding the fistula (Figure 6 and Figure 7). The excluded segment was then visualized by injection of contrast through the surgical drain under fluoroscopy (Figure 8). A 19-gauge FNA needle was used to puncture the orphan stomach from the remnant (Figure 9), followed by a guidewire (Figure 10, Figure 11 and Figure 12). The tract was dilated up to 6 mm with a balloon (Figure 13, Figure 14, Figure 15 and Figure 16), and a 8 mm × 40 mm covered metal biliary stent was placed across the divide (Figure 17 and Figure 18). Then, a 7Fr × 4 cm double pigtail stent was placed inside this to facilitate drainage (Figure 19 and Figure 20). The gastrostomy was replaced, and the surgical drain was removed.

Six weeks later, repeat retrograde endoscopy was performed, and the covered stent and pigtail were removed. A 5-mm gastrogastrostomy was noted. Two 7 Fr × 4 cm double pigtail stents were placed to maintain patency of the therapeutic gastro-gastric fistula. The remnant gastrostomy was removed.

One month later, the patient reported a 12-pound weight gain and was feeling well. CT scan showed stents in place and no collections (Figure 21).

Over the subsequent year, she was referred to colleagues in bariatric medicine due to mild weight regain and patient concerns about excessive weight regain. She had a follow up CT scan after one year that showed the pigtail drains were still in place (Figure 22). She has been referred back to GI to discuss retrograde endoscopy through the biliopancreatic limb and stent removal.

## 3. Discussion

An “orphaned” stomach segment is an unusual complication after revisional gastric surgery that is not well represented in the surgical literature. One similar case was reported after gastric bypass reversal [19], where a non-partitioned gastric bypass was reversed by resection of the gastrojejunostomy and creation of a gastrogastrostomy. In this case, postoperative leukocytosis and symptoms prompted a CT scan, which revealed a distended isolated gastric segment between staple lines. The surgeons had a high index of suspicion and were able to perform antegrade endoscopy with endoscopic ultrasound (EUS) to facilitate trans-gastric stenting, which resolved the issue.

In the present case report, we have shown that an isolated stomach segment may also occur after revision of a prior bariatric procedure. Both vertical banded gastroplasty and gastric bypass may require revision due to complications or weight regain. Vertical banded gastroplasty can have band erosion or staple line breakdown. RGB can have recurring marginal ulcers, gastrojejunal strictures, pouch dilation, or gastro-gastric fistula from staple line dehiscence. It is important to note that, before the development of the current generation of endostaplers, which divide tissue between rows of staples, bariatric procedures utilized multi-row partitioning staplers that did not divide and separate the stomach segments. Sometimes, such staple lines can recanalize, making revisional procedures necessary.

In reoperative cases, the presence of scar and the nature of staple line healing make it very difficult for a surgeon to recognize the location of residual areas of intact pre-existing staple lines. Therefore, during revision gastroplasty, it is possible for a new staple line to inadvertently cross an existing length of intact prior staple line, such that a segment of stomach can have limited or no continuity with the gastric outflow channel.

Surgical strategies to limit this risk include use of intraoperative endoscopy, or, if laparoscopic visualization does not allow appreciation of the course of prior staple lines, use of laparotomy to allow tactile benefits. In addition, when one is operating to re-establish gastric restriction after an incomplete staple line dehiscence in the setting of a previously non-divided gastroplasty, it may be prudent to open the stomach and internally divide any residual areas of intact partition (using a linear stapler) prior to forming a new divided gastroplasty.

Use of directed transgastric stents is increasingly common for other indications. Mature pancreatic pseudocysts can be internally drained through the posterior stomach with endoscopic and ultrasound guidance [20,21,22,23]. After RGB, gastro-gastric stents have been used to allow antegrade access to the duodenum and biliary system for diagnostic and therapeutic procedures [20,24,25]. In addition, in cases of gastric or duodenal obstruction, transgastric stents have been placed for gastro-enterostomy [26,27,28]. In all of these situations, EUS is very helpful to visualize the target organ.

In the present case, EUS was not possible due to the use of a gastrostomy site for access to the remnant stomach. Instead, sinogram under fluoroscopy allowed visualization of the target for stenting and assessment of the tissue bridge to be traversed. Use of retrograde endoscopy via a gastrostomy site has been conducted before [29,30,31,32]. In such cases, it is important to allow the gastrostomy site to mature, such that the stomach is adherent to the anterior abdominal wall. Sufficient gastric fixation usually requires the tube be in place at least four weeks, or longer in malnourished or diabetic patients [33,34,35].

Recurring abscesses after revisional gastric surgery without a leak from the alimentary path should raise suspicion of a gastric fistula arising from an unrecognized leak from the bypassed gastric remnant or, in rare cases, from an “orphaned” segment of stomach. In the case of an isolated stomach segment separate from the proper gastric remnant, multidisciplinary evaluation with interventional radiology, gastroenterology, and bariatric surgery providers can help make the diagnosis and allow consideration of advanced endoscopic techniques for internal drainage without the need for major gastric resection.

## Figures and Tables

**Figure 1 jcm-11-07487-f001:**
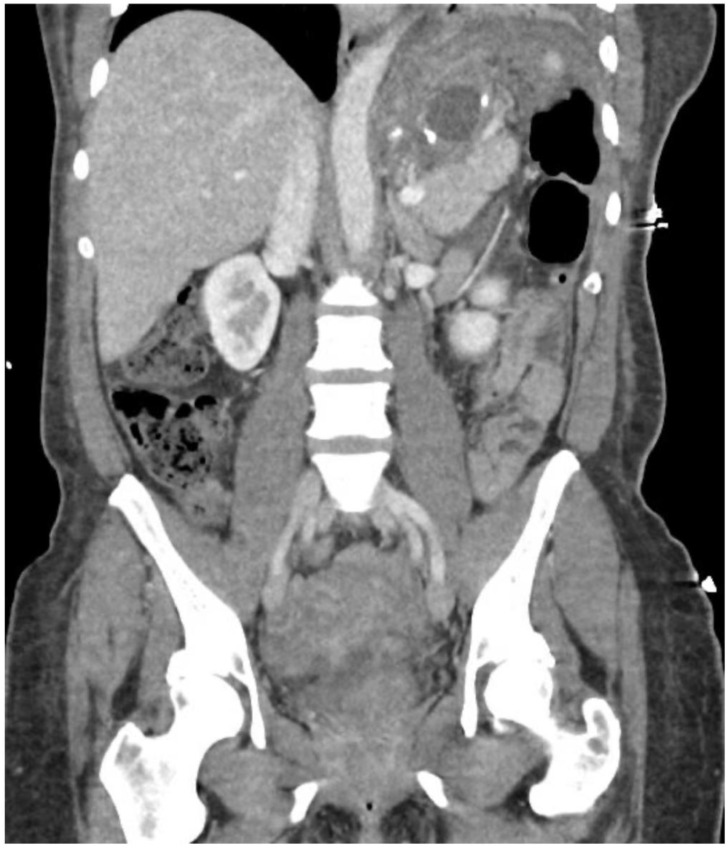
Initial CT showing LUQ abscess.

**Figure 2 jcm-11-07487-f002:**
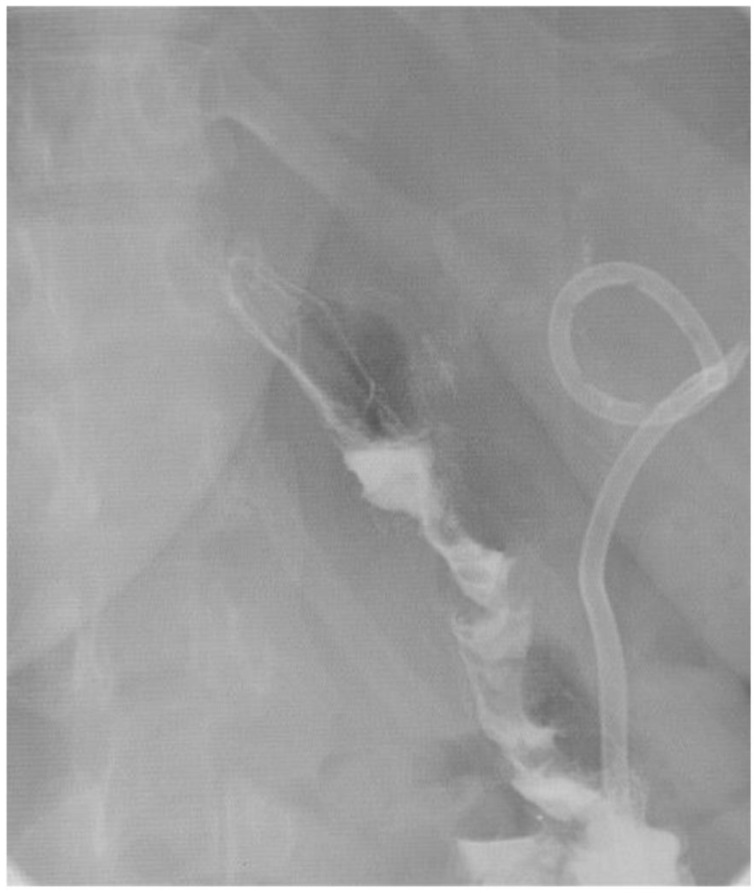
UGI series showing no leak from proximal stomach.

**Figure 3 jcm-11-07487-f003:**
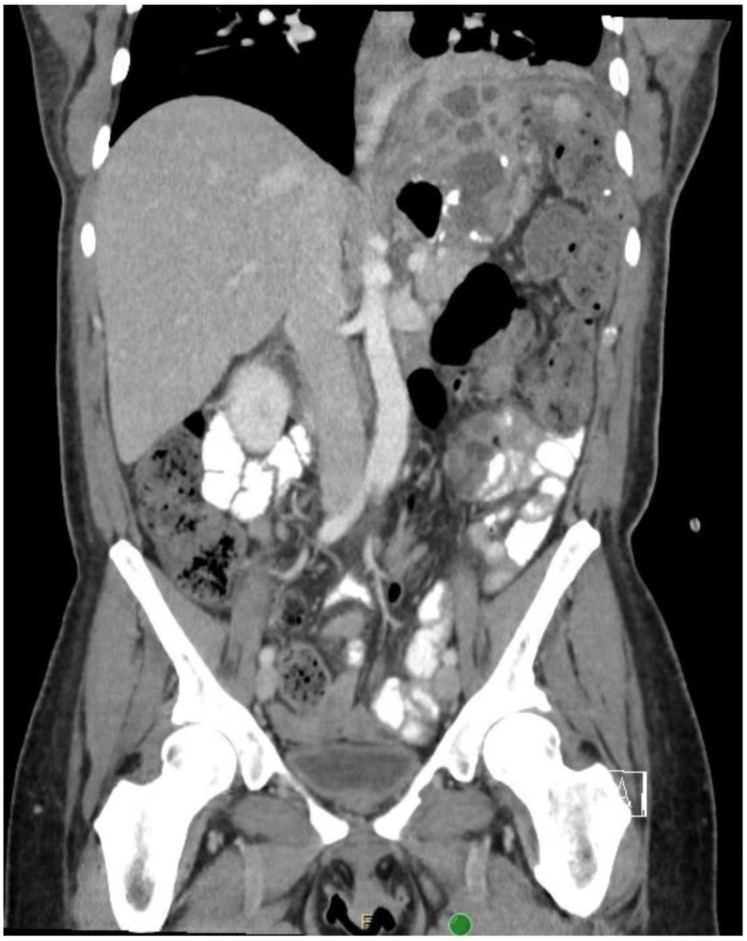
CT showing multiloculated abscess and suspected orphaned stomach segment.

**Figure 4 jcm-11-07487-f004:**
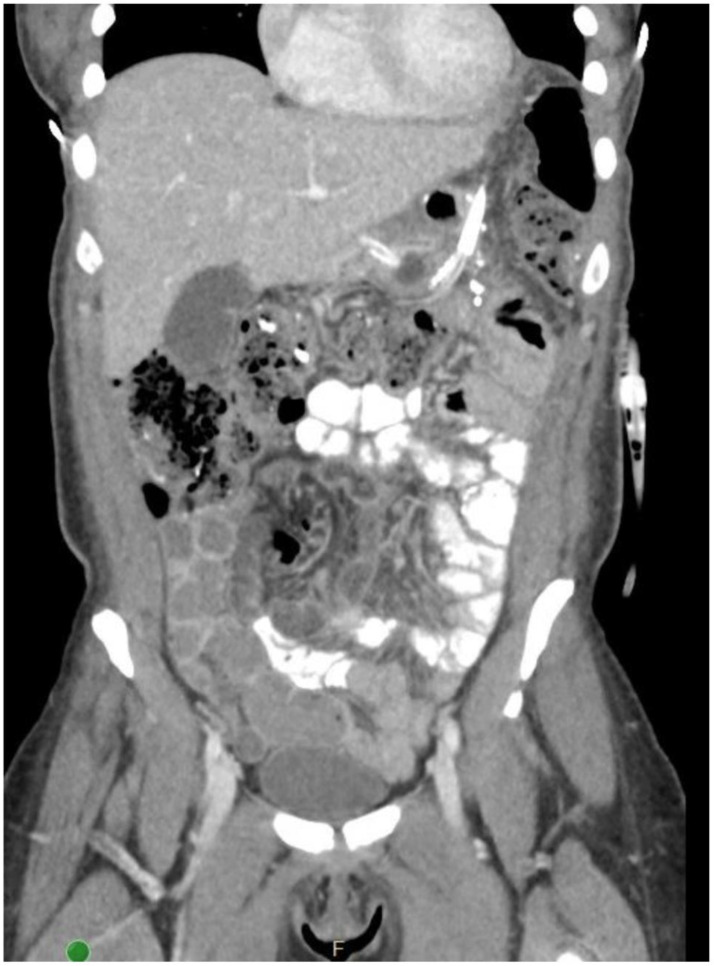
Abscess drained and remnant G-tube placed (not visible) after laparoscopic surgery.

**Figure 5 jcm-11-07487-f005:**
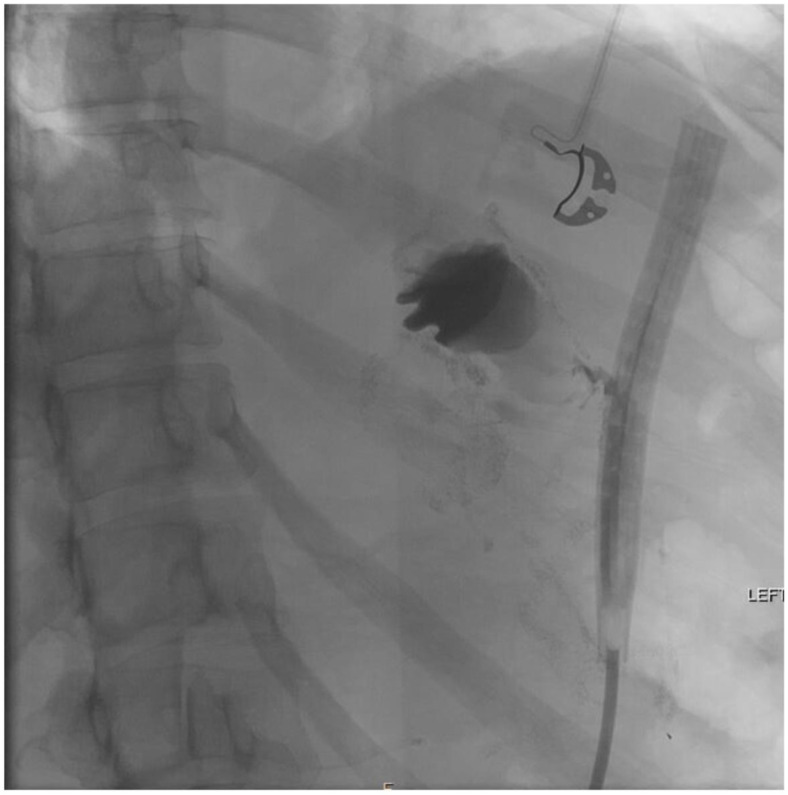
Sinogram showing fistula to orphaned stomach segment.

**Figure 6 jcm-11-07487-f006:**
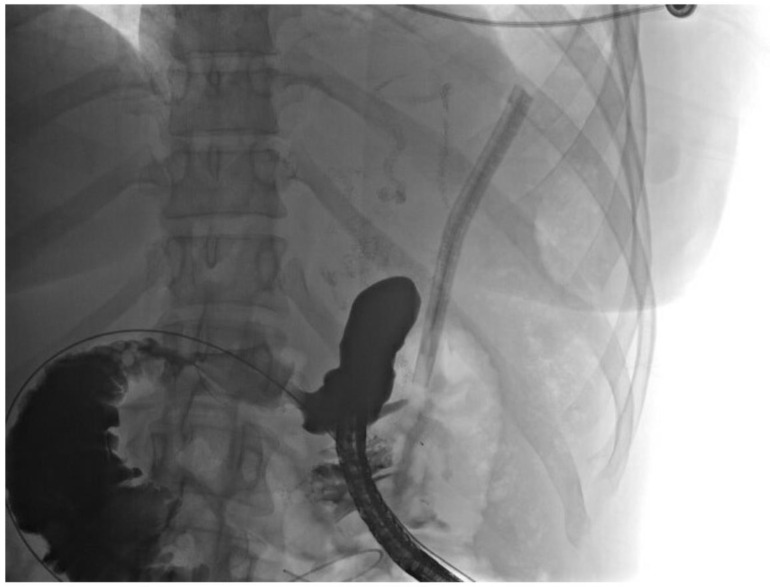
Endoscopic access to remnant stomach via mature gastrostomy site.

**Figure 7 jcm-11-07487-f007:**
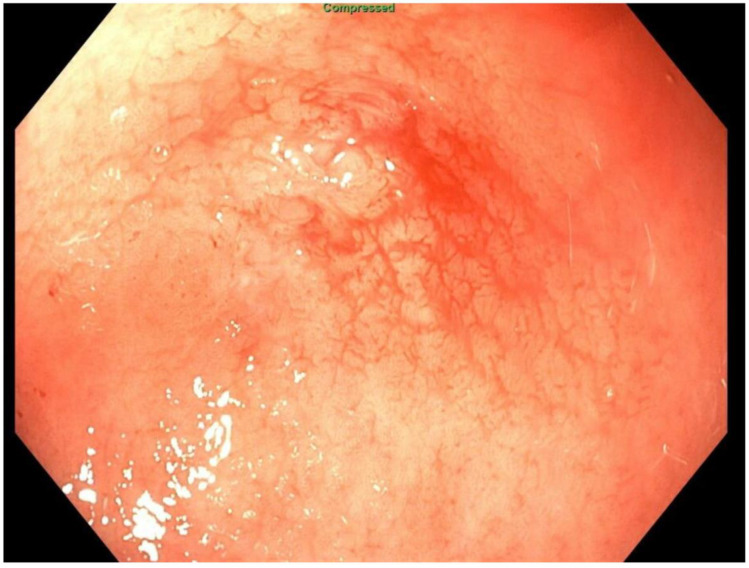
Endoscopic view of blind proximal end of gastric remnant.

**Figure 8 jcm-11-07487-f008:**
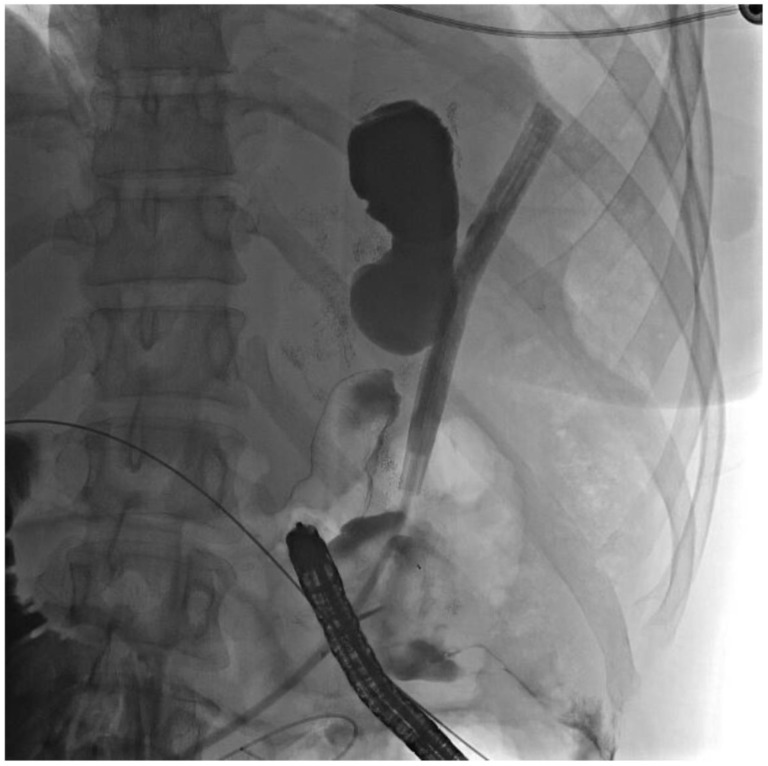
Sinogram during retrograde endoscopy to localize orphaned segment of stomach.

**Figure 9 jcm-11-07487-f009:**
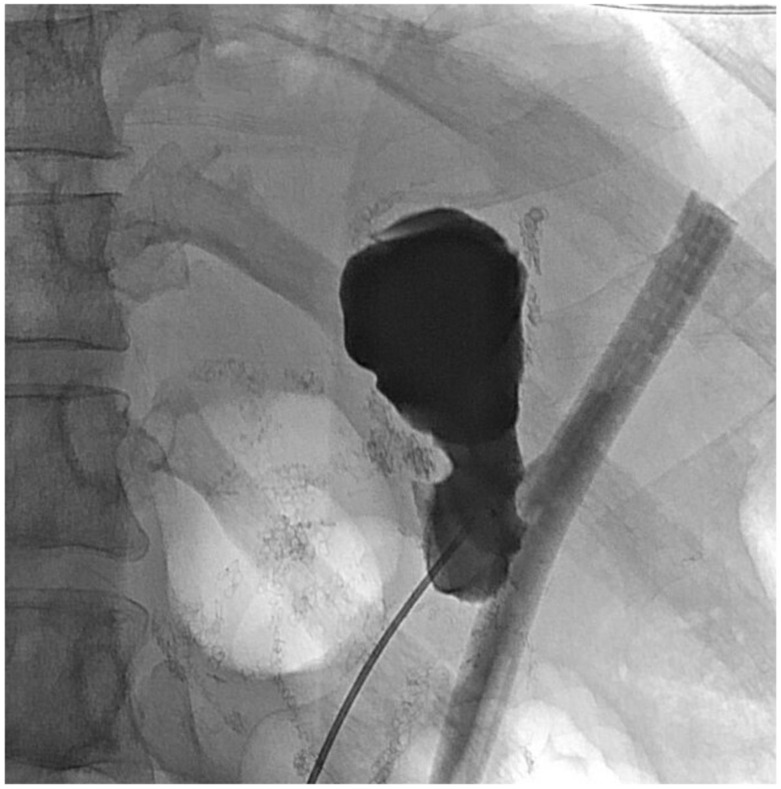
Needle access into orphaned stomach segment.

**Figure 10 jcm-11-07487-f010:**
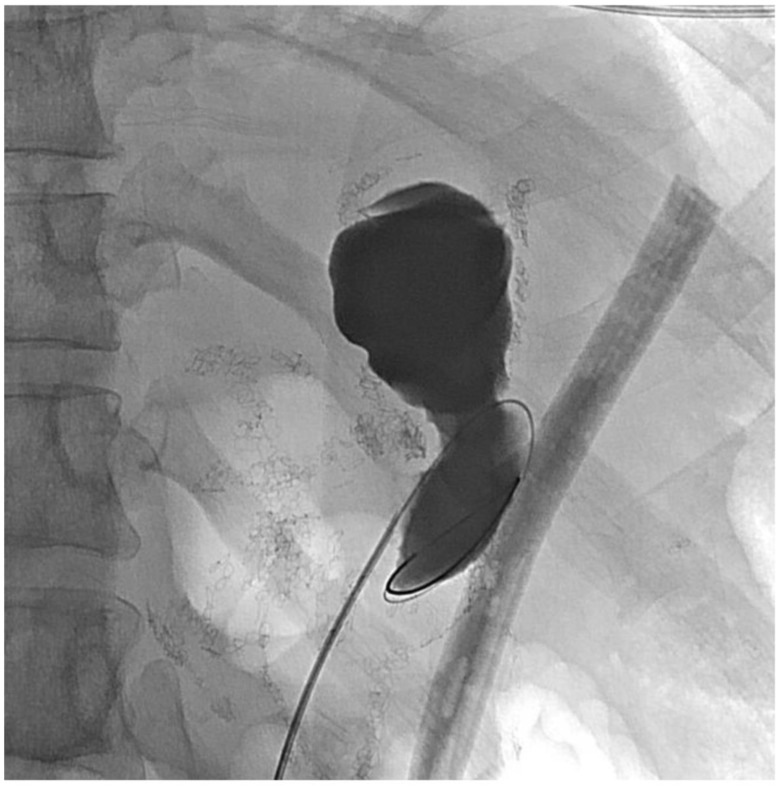
Wire across into orphaned stomach segment (fluoroscopic view).

**Figure 11 jcm-11-07487-f011:**
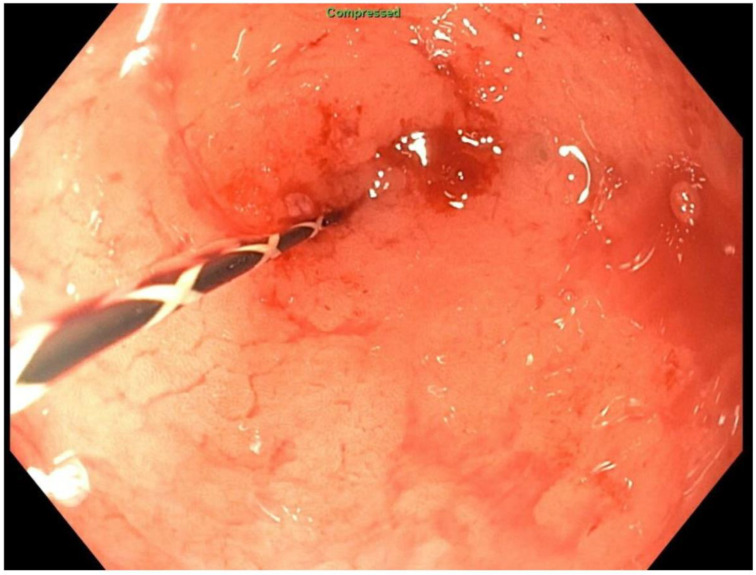
Wire through gastric remnant into orphaned stomach (endoscopic view).

**Figure 12 jcm-11-07487-f012:**
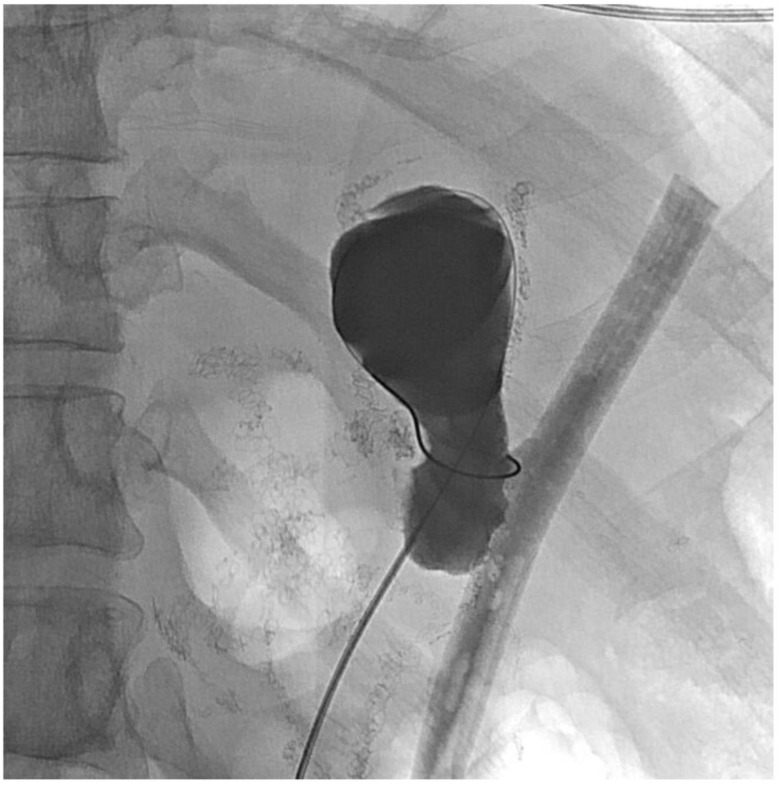
Wire definitely in orphaned stomach segment.

**Figure 13 jcm-11-07487-f013:**
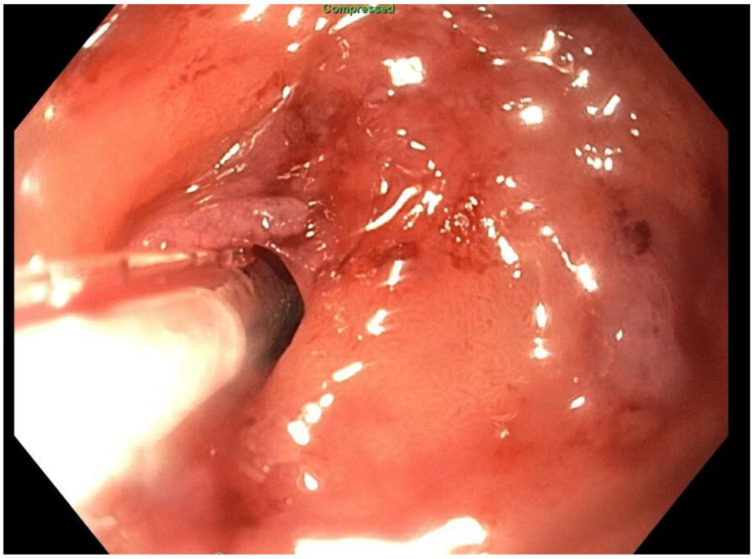
Balloon dilation across the divide.

**Figure 14 jcm-11-07487-f014:**
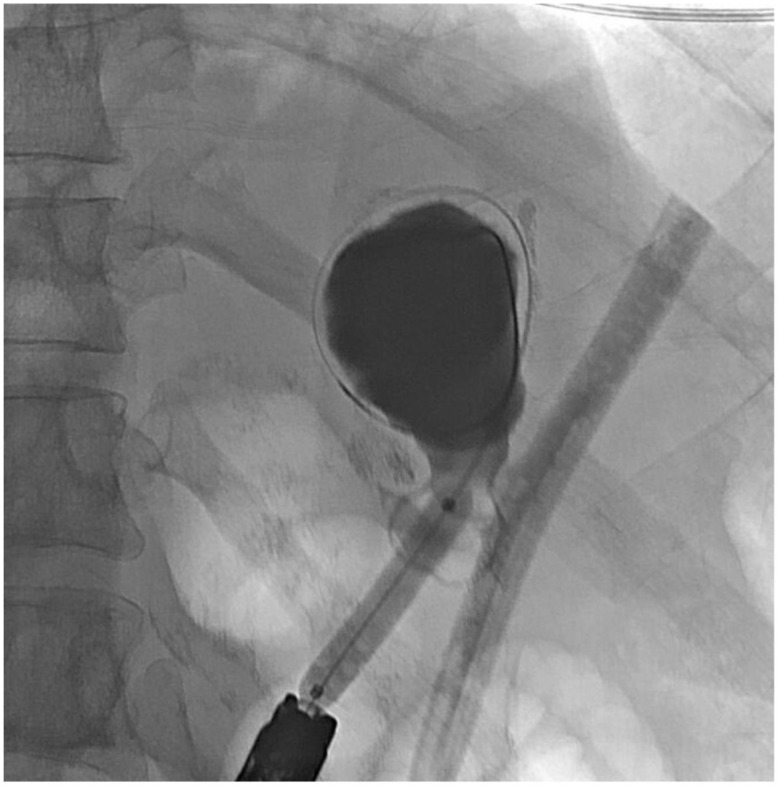
Balloon dilation (fluoroscopic view).

**Figure 15 jcm-11-07487-f015:**
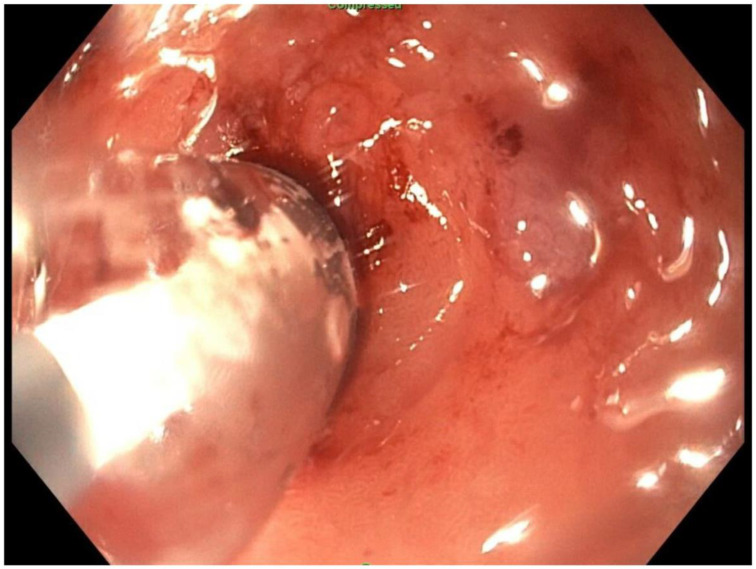
Dilating the balloon.

**Figure 16 jcm-11-07487-f016:**
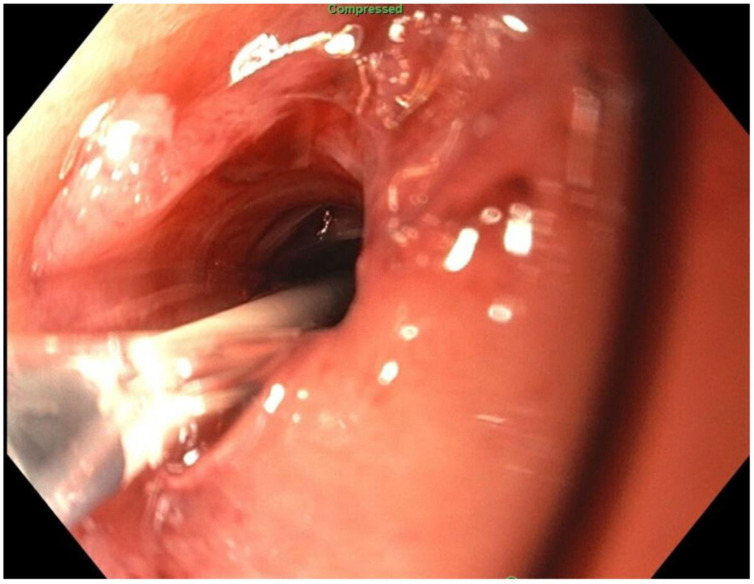
Appearance after dilating.

**Figure 17 jcm-11-07487-f017:**
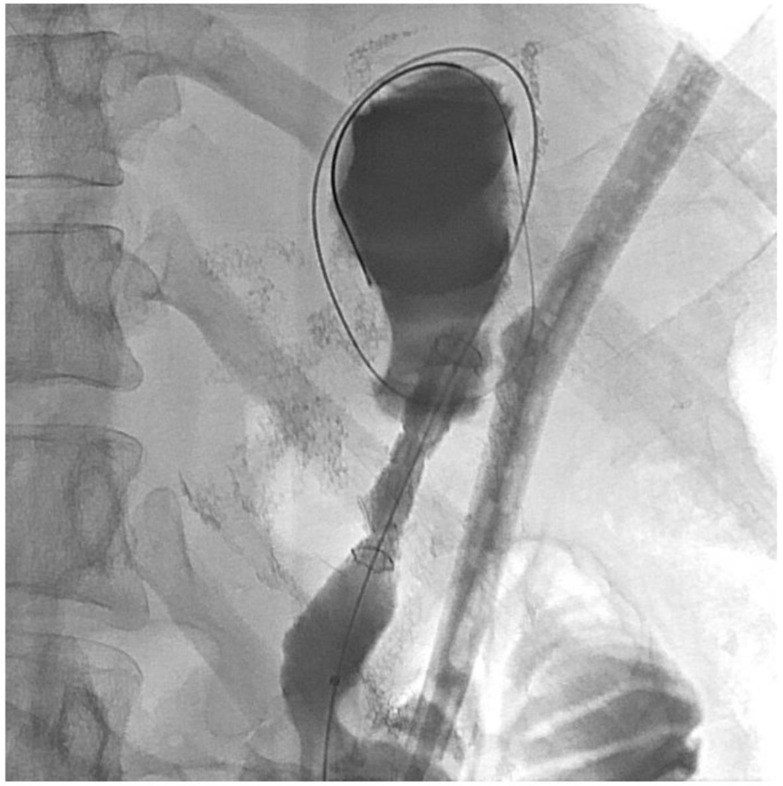
Covered stent insertion (fluoroscopic view).

**Figure 18 jcm-11-07487-f018:**
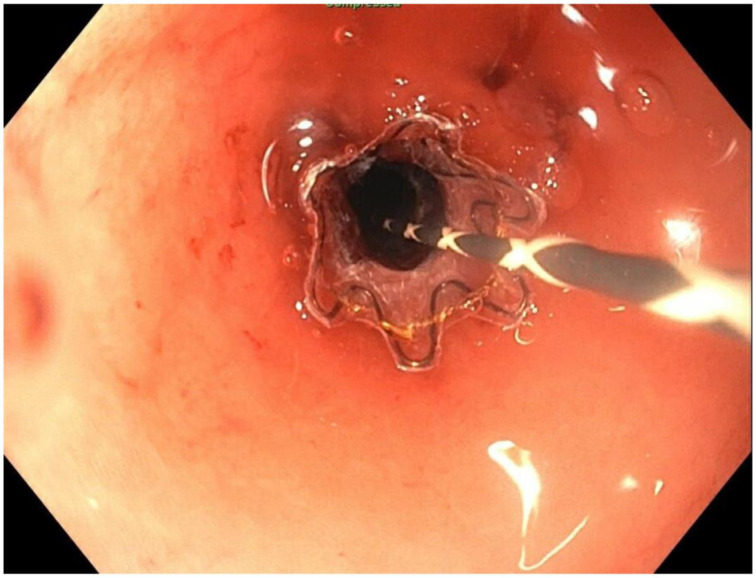
Covered stent insertion (endoscopic view).

**Figure 19 jcm-11-07487-f019:**
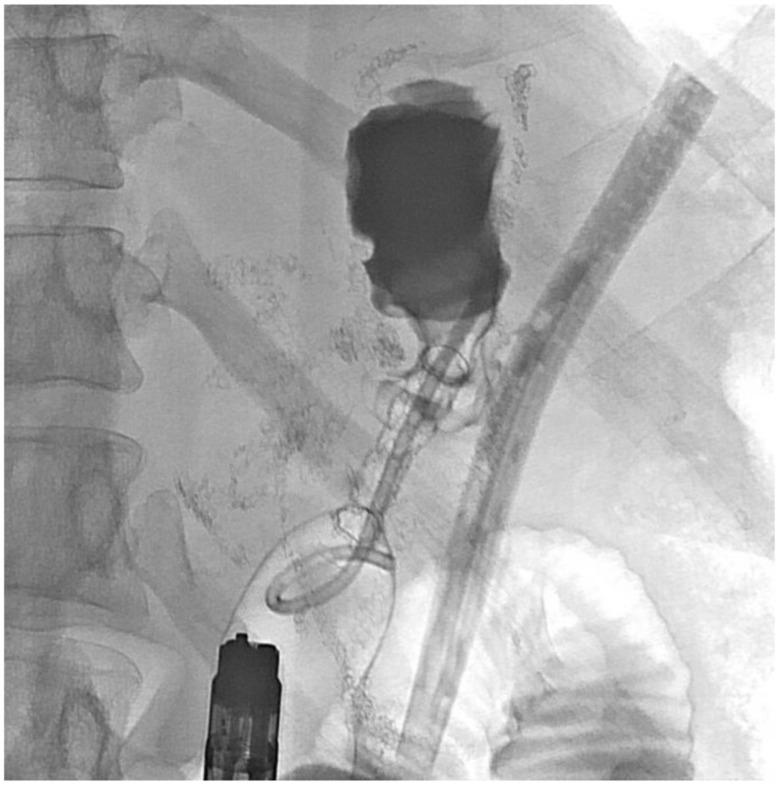
Pigtail stent through covered stent (fluoroscopic view).

**Figure 20 jcm-11-07487-f020:**
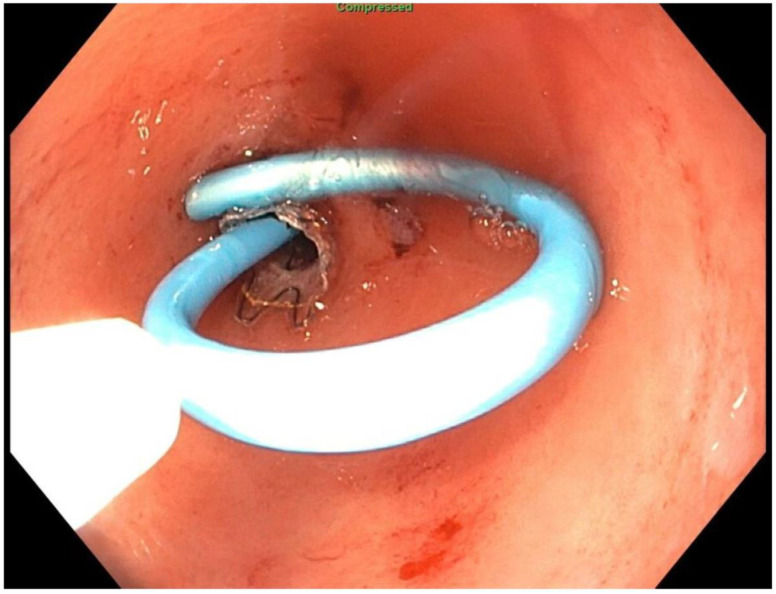
Pigtail stent through covered stent (endoscopic view).

**Figure 21 jcm-11-07487-f021:**
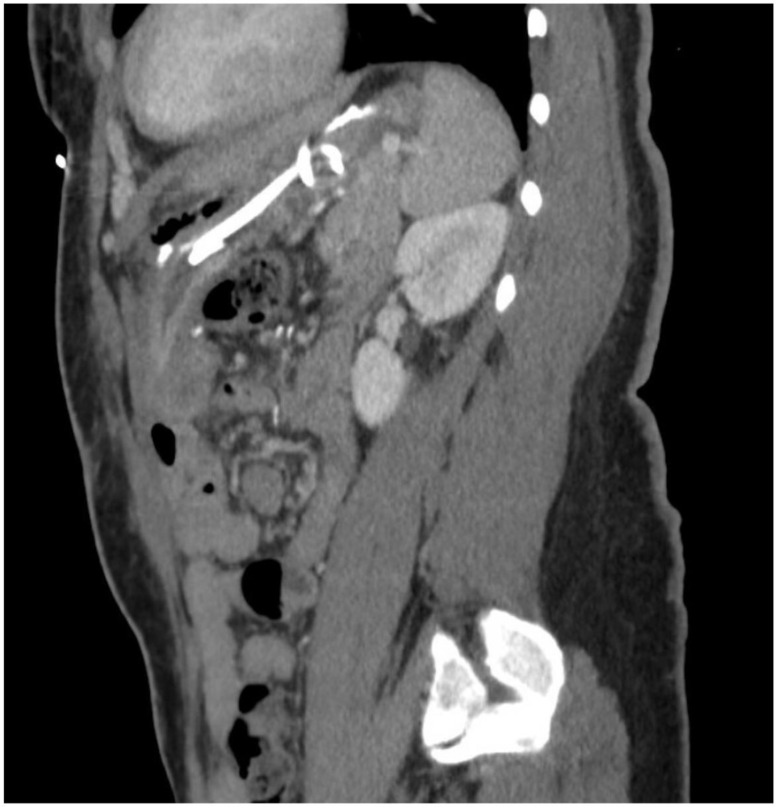
Internal pigtail stents seen on lateral view.

**Figure 22 jcm-11-07487-f022:**
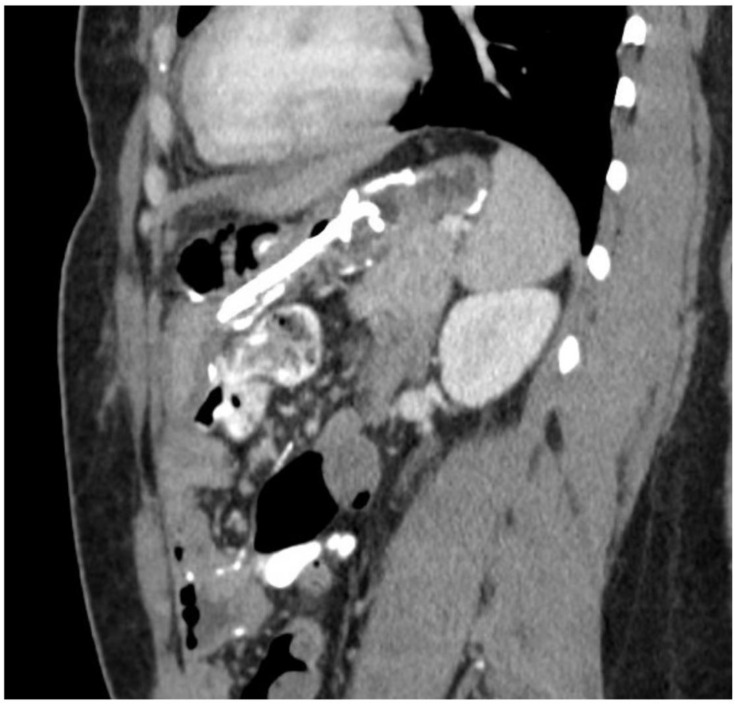
Pigtail stents remain in place 1 year later, again seen on lateral view.

## Data Availability

Not applicable.

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
