# Peer review of "“Orphaned” Stomach—An Infrequent Complication of Gastric Bypass Revision"

_jcm, 2022, doi:10.3390/jcm11247487_

Round 1

Reviewer 1 Report

In this manuscript, the authors report on the case of an obese patient who initially underwent a laparoscopic roux-en-Y gastric bypass and after about 5 years, due to weight regain, underwent a laparoscopic revision, consisting in the reduction of her gastric pouch and redo of gastrojejunostomy. 

Early after this second operation, the patient developed (not clear if one or more) intraperitoneal collection(s) requiring a surgical exploration. Intraoperatively, such collection were diagnosed as excluded segment of the remnant stomach  and managed in an unclear way. 

Thereafter, due to collection recurrence in the left upper abdominal quadrant, the patient was readmitted to the hospital and underwent collection percutaneous drainage with clinical improvement, however after the drain removal (3 weeks later) a new recurrence of the abdominal collections was observed. THe patient underwent a new operation, consisting in adhesiolisis and gastrostomy to decompress the remnant stomach.  Subsequent multistep endoscopic (via gastrostomy) and interventional radiologic procedures allowed the successful internal drain of the excluded gastric "orphaned" stomach in the remnant stomach. 

Despite overall well written and concerning the complex successful management of a rare (and probably misdiagnosed) complication of bariatric surgery, this manuscript deserves some comments. 

- the manuscript contains few grammatical errors (line 67: separate should be separated) need to be corrected.

- I could not see the figures cited in the manuscript: please send me a manuscript version with the pictures embedded.

- However, there are too many  figures in the manuscript: the authors should either select the most useful figures to show or assemble more figures  in a single image.

- Lines 52-53: how was the segment of excluded stomach treated? was it just left there?

- In the discussion (paragraphs 2, 3), the authors describe the hypothetical mechanisms leading to the development of the orphaned stomach: I find this section quite difficult to understand, I suggest the authors to improve it. I also believe that adding an illustration describing the suspected mechanism which led to the development of the "orphaned" stomach may increase the manuscript clearness. 

- an additional illustration may be added in order to better clarify the endoscopic/radiologic interventional management of the orphaned stomach may further increase the manuscript understandability. 

Reviewer 2 Report

Dear Authors,

With pleasure I read your interesting case report and review of the literature on the topic of the "Orphaned" stomach. The report is well written and covers a topic that should interest every bariatric surgeon performing redo surgery.  

Round 2

Reviewer 1 Report

I believe that the authors have correctly responded to my queries and that the manuscript has improved following recent changes.
